# Balancing Graph Embedding Smoothness in Self-supervised Learning via Information-Theoretic Decomposition

## Abstract

In the graph domain, SSL has garnered significant attention, particularly in employing Graph Neural Networks (GNNs) with pretext tasks originally designed for other domains, such as contrastive learning and feature reconstruction. However, it remains uncertain whether these methods effectively reflect essential graph properties, such as representation similarity with its neighbors. We observe that existing methods position opposite ends of a spectrum driven by the graph embedding smoothness, with each end corresponding to outperformance on specific downstream tasks. Further insights suggest that balancing between the extremes can lead to improved performance across a wider range of downstream tasks. To find the balance respective to the graph embedding smoothness, we decompose the SSL objective into three terms, which are derived by incorporating the neighbor representation variable through the lens of information theory. A framework, **BSG** (**B**alancing **S**moothness in **G**raph SSL), introduces novel loss functions designed to supplement the representation quality in graph-based SSL by optimizing the derived three terms: neighbor loss, minimal loss, and divergence loss. We present a rigorous theoretical analysis of the effects of these loss functions, highlighting their significance from both the SSL and graph smoothness perspectives. Extensive experiments on multiple real-world datasets across node classification and link prediction consistently demonstrate that BSG achieves state-of-the-art performance, outperforming existing methods. Our implementation code is available at https://anonymous.4open.science/r/BSG-2025/.

## CCS Concepts

• **Computing methodologies** → **Learning latent representations**; • **Information systems** → **Data mining**.

## Keywords

Self-supervised Learning, Graph Neural Network, Oversmoothing

**ACM Reference Format:**
Anonymous Author(s). 2018. Balancing Graph Embedding Smoothness in Self-supervised Learning via Information-Theoretic Decomposition. In *Proceedings of Make sure to enter the correct conference title from your rights confirmation emai (Conference acronym 'XX)*. ACM, New York, NY, USA, 13 pages. https://doi.org/XXXXXXX.XXXXXXX

## 1 Introduction

Self-supervised learning (SSL) is a label-free paradigm that significantly reduces the cost of obtaining labels. SSL uncovers underlying patterns in unannotated data by defining self-tailored tasks [10]. The primary objective of SSL is to produce sufficient and minimal representations with the self-supervised tasks, which the learned representations are easily adapted into various downstream tasks [29, 36]. The field has advanced through various techniques, including reconstruction-based methods and contrastive learning methods [20]. For instance, reconstruction-based methods first mask the portion of the input and predict masked inputs using contextual information [4–6, 52]. Contrastive learning improves representations by ensuring closer alignment of positive samples and increased separation of negative samples [11, 14].

Graph SSL research is burgeoning, primarily focusing on contrastive learning and reconstruction-based methods. Contrastive methods for graphs typically generate alternative views by augmenting graph inputs through feature masking [47, 48]. Masked graph auto-encoders (MGAE) mask graph features and reconstruct the masked parts [8, 18]. Most contrastive learning and node feature reconstruction-based models relatively outperform in node classification task, and edge reconstruction-based models achieve high performance in link prediction downstream task [28]. However, both contrastive and reconstruction-based methods have their origins in other domains [21, 39], suggesting that they may not fully consider the unique structure of inter-connected graph data. Most methodologies simply utilize Graph Neural Networks (GNNs) as encoders [18, 28, 32] to adapt these techniques to graph data without fully exploring the intrinsic properties of graphs.

To assess whether existing studies capture the underlying properties of graphs, we conducted an experiment to analyze how the similarity between learned node representations and their neighbors influences the performance of graph SSL models. Specifically, we examined the graph embedding smoothness [44] produced by existing graph-based SSL methods. For instance, node feature reconstruction models [18, 37] estimate masked features by relying on their neighbors' unmasked representations, which results in smoother representations. Interestingly, as shown in Figure 1, feature reconstruction (FR) [18] and contrastive learning (CL) method [48], which outperform in node classification tasks, tend to produce smoother representations compared to the edge reconstruction (ER) method [18], which outperform in link prediction tasks. A further observation from the figure suggests that achieving a balance in similarity with neighboring nodes leads to consistent outperformance across various downstream tasks.

To acquire the balanced representation, we first delve into Information Theory [3] by adding a new variable representing the neighboring node to the original self-supervised learning objective, which is maximizing the mutual information between the learned representation and their respective self-supervised signals [34].

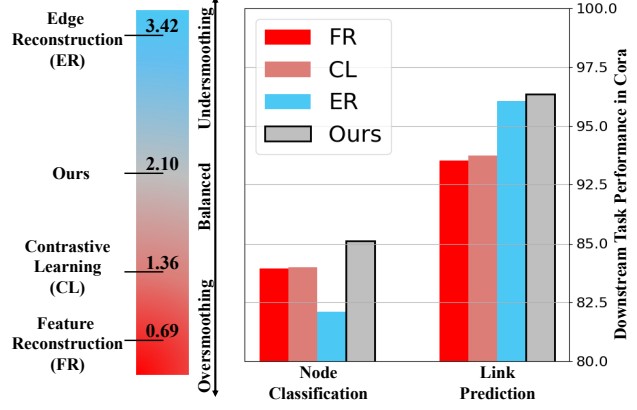

**Figure 1: A comparison of existing graph SSL baselines, with colors indicating the smoothness of representations on a log-scale (red is more smoothed) and the height of each bar corresponds to the performance of downstream tasks (accuracy). The smoothness is acquired by the distance between node representation and their neighbor's mean representation.**

To fully leverage graph smoothness within the SSL domain, we introduce **BSG** (**B**alancing **S**moothness in **G**raph SSL), which incorporates novel loss functions designed to effectively capture the key terms in information theory due to an additional neighbor variable. Our approach proposes three loss functions: neighbor loss, divergence loss, and minimal loss. The significance of each term can be interpreted in both the SSL and graph perspectives. The neighbor loss explicitly increases the mutual information between the node and its neighbors by increasing the smoothness of the graph, and it relates to the sufficient representation in SSL. The divergence loss function has a trade-off relation with the first loss to alleviate the risk of falling into a trivial solution of identical representation between the learned representation and its neighbors. From the perspective of graphs, the divergence loss aligns with alleviating the oversmoothing problem. Furthermore, to achieve consistent performance on various downstream tasks, discarding task-irrelevant information is essential in the graph domain. The minimal loss function effectively discards task-irrelevant information, resulting in a minimal representation in SSL. BSG, which incorporates the newly proposed loss functions, generates representations that are more balanced, as shown in Figure 1.

Extensive experimental evaluations on various benchmarks, including node classification and link prediction, demonstrate the effectiveness of BSG. Our method shows promising results across all experimental settings compared to the leading baselines.

## 2 Related Work

### 2.1 Self-supervised Learning on Graphs

We categorize the existing studies into three: auto-encoding, contrastive learning, and mask modeling.

*2.1.1 Auto-encoding.* The pioneering work of GAE [15] encodes the graph into latent representation and decodes it to reconstruct the original graph topology or features. However, these methodologies solely focusing on reconstructing direct topology overemphasize the structural proximity information that hinders specific downstream tasks like node classification tasks [18], which is partially not adequate for the SSL objective.

*2.1.2 Contrastive Learning.* Contrastive learning, another key approach, creates alternative graph views or compares them with the entire graph to identify positive and negative samples, as seen in DGI [35] and InfoGraph [31]. Most methods learn invariant representations that remain robust to simple augmentations, which can be interpreted as a reliance on consistent neighbors. As a result, they are susceptible to the potential risk of oversmoothing.

*2.1.3 Mask Modeling.* Masked Graph Auto-Encoder (MGAE) techniques mask parts of the input, including edges, features, and latent space, and reconstruct hidden masked elements [8, 18, 28]. For example, MaskGAE [18] masks the edge (path). By setting an objective to reconstruct the masked edge (path), it highly outperforms on the related downstream link prediction task. Most methods adopt a high ratio of edge masks. This results in discerning the features of nodes with only the partial information of their neighbors, indicating the representations to be undersmoothed.

### 2.2 Smoothness of Graph Embeddings

GNN and its similar neighbor aggregation methods have the advantage of sharing information, making a more smoother representation. However, they often fall into a degenerated solution in which every representation in the graph becomes similar, and this phenomenon is called oversmoothing [19]. Since this phenomenon is a fatal factor to underperformance, a lot of study is proposed to alleviate this problem by introducing residual connection [2, 17], dropping edges randomly [25], normalization [50], and regularization [1]. However, as noted in [13], it is crucial to identify the optimal balance since a certain amount of graph smoothing enhances performance without leading to oversmoothing.

Similarly, there are some studies that strive to maintain the global feature smoothness in the SSL domain [12, 37]. For example, a recent study [37] points out the lack of uniformity in the global representations by adopting the feature reconstruction SSL objective and addresses the challenge through explicit regularization. However, global feature smoothness is not particularly related to the inherent structure of graphs, as these objectives primarily emphasize the overall smoothness at the representation level rather than regarding the smoothness relative to neighboring nodes. Unlike previous methods, the representations are oversmoothed or undersmoothed, our work aims to find a balance between the representations with their neighbors in the SSL domain to achieve robust performance on various downstream tasks.

## 3 Preliminary

### 3.1 Notations

We denote the random variable for the input as $\mathbf{X}$, and the raw input as $\mathbf{x}$. Similarly, the random variable and the outcome of the latent representation are represented as $\mathbf{Z_X}$ and $\mathbf{z_x}$, respectively. For the random variables of the self-supervision signal and downstream task signal, we denote $\mathbf{S}$ and $\mathbf{Y}$, respectively. We note $I(\mathbf{X}; \mathbf{Y})$

and $I(X; Y|Z)$ to represent the mutual information and conditional mutual information for random variables $X, Y$, and $Z$, respectively. $H(X)$ and $H(X|Y)$ means the entropy and conditional entropy.

We consider a graph $G(A, x)$. Here, $A \in \{0,1\}^{N \times N}$ denotes adjacency matrix with $N$ indicating the number of nodes, and $x \in \mathbb{R}^{N \times D_{node}}$ represents node features matrix. $\mathcal{V}$ and $\mathcal{E}$ denotes set of nodes and edges in $G$ respectively. The edge mask $M \in \mathbb{R}^{N \times N}$ is applied to the original adjacency matrix, and its masked graph is represented as $\hat{G}(M \otimes A, x)$, where $\otimes$ is the Hadamard product. The encoded latent representation by processing the graph $\hat{G}$ with a Graph Convolutiona Network (GCN) [16] encoder is represented as $z_x \in \mathbb{R}^{N \times D_{emb}}$.

## 3.2 Information Bottleneck for SSL

The Information Bottleneck (IB) [33] for the supervised setting aims to learn latent representations with high mutual information with the downstream task while containing minimal redundancy with the given input, and can be formalized as:

$$\max_{\theta} I(Z_X; Y) - \beta I(Z_X; X), \tag{1}$$

where $\theta$ is the parameter to optimize the equation, $\beta$ is a hyperparameter to control the two terms, and $Y$ is the random variable of its downstream task label. The information bottleneck strongly aligns with the sufficient and minimal representation, which $I(Z_X; Y)$ makes the representation to satisfy the sufficient condition, and $(-\beta I(Z_X; X))$ is related to the minimal condition.

For the self-supervised learning (SSL) setting, most methods indirectly optimize Equation (1) by maximizing $I(Z_X; S)$ and minimizing $I(Z_X; X)$ since the self-supervised setting utilizes a pretext task label $S$ instead of $Y$. Therefore, the minimal and sufficient representations for SSL can be formalized by assuming the multi-view assumption [29, 30, 36], which is expressed as:

**Definition 1.** *(Minimal and Sufficient Representations for SSL). Let $Z_X^{ssl}$ be the sufficient representation and $Z_X^{ssl\_min}$ be the minimal representations for SSL:*

$$Z_X^{ssl} = \arg\max_{Z_X} I(Z_X; S),$$
$$Z_X^{ssl\_min} = \arg\min_{Z_X} H(Z_X|S), \quad s.t. \quad I(Z_X; S) \quad is\ maximized. \tag{2}$$

The multi-view assumption assumes that $I(Z_X; Y)$ and $I(Z_X; S)$ are positively correlated. The contrastive learning is directly increasing the mutual information between $Z_X$ and $S$ since they are treated as positive samples. In the case of reconstruction, it is minimizing the entropy $H(S|Z_X)$. Since $I(Z_X; S)$ can be decomposed into $H(S) - H(S|Z_X)$, minimizing $H(S|Z_X)$ has identical effect to increasing $I(Z_X; S)$ [34]. Therefore, the objective of SSL, including contrastive learning and reconstruction through masking, aligns with sufficient representation.

## 4 Methodology

This section details the architecture of BSG. First, we add an additional variable indicating the random variable of neighbor representation $Z_N$ to the SSL objective, maximizing $I(Z_X; S)$. Then, three additional terms can be derived by incorporating $Z_N$, leading us to propose three corresponding loss functions that align with

each term. The remaining section introduces the new SSL objective for the graph, an overview of BSG, details of each loss function, and its theoretical analysis.

## 4.1 SSL Objective for Graph

Conventional SSL's objective is to maximize the mutual information between $Z_X$ and $S$. To fully consider the inter-connected graph structure, we need to consider its neighbor representations. The SSL objective can be easily decomposed by adding an additional variable $Z_N$, which indicates the random variable of neighbor representation. Therefore, the equation can be decomposed as:

$$I(Z_X; S) = I(Z_X; Z_N) - I(Z_X; Z_N|S) + I(Z_X; S|Z_N). \tag{3}$$

The first term of the equation, which indicates the mutual information of $Z_X$ and $Z_N$, is increased when adopting a GNN structure. Therefore, existing GNN-based SSL methodologies for graphs can be treated as increasing $I(Z_X; S)$ with implicitly increasing $I(Z_X; Z_N)$ by aggregating messages from its neighbors. However, as shown in Figure 1, differences in neighbor similarity due to variations in $S$ impact the downstream task performance, indicating that addressing the additional terms $I(Z_X; Z_N|S)$ and $I(Z_X; S|Z_N)$ is essential to fully maximize $I(Z_X; S)$ in graph-based analyses. To address each term, we reformulate the equation by transforming the mutual information to the entropy term like:

$$I(Z_X; Z_N|S) = H(Z_X|S) - H(Z_X|Z_N, S),$$
$$I(Z_X; S|Z_N) = H(Z_X|Z_N) - H(Z_X|Z_N, S). \tag{4}$$

By exchanging the terms in Equation (3) into Equation (4), the resultant term can be induced like:

$$I(Z_X; S) = I(Z_X; Z_N) - H(Z_X|S) + H(Z_X|Z_N). \tag{5}$$

The objective of SSL, which is maximizing $I(Z_X; S)$, can be interpreted as maximizing $I(Z_X; Z_N)$, minimizing $H(Z_X|S)$, and maximizing $H(Z_X|Z_N)$. Finally, the objective can be expressed as:

$$\max I(Z_X; S) \approx \max(I(Z_X; S) + \lambda_1 I(Z_X; Z_N) \\ -\lambda_2 H(Z_X|S) + \lambda_3 H(Z_X|Z_N)), \tag{6}$$

where $\lambda_1, \lambda_2, \lambda_3$ are positive hyperparameters to control the effect of each term. The details of each term will be discussed in the following sections.

## 4.2 Overview

The overall framework, including the process of acquiring the necessary representations and the proposed loss function, is illustrated in Figure 2. Besides $z_x$, we additionally acquire $z_s$, and $z_{neigh}$, which are the representations obtained by transforming the visible and masked edge, and the mean value of its neighbor representation, respectively. From the acquired representations, three loss functions can be defined to capture the underlying graph features by balancing the graph embedding smoothness of representations.

## 4.3 Neighbor Loss is Maximizing $I(Z_X; Z_N)$

$I(Z_X; Z_N)$ is explicitly increased by maximizing the dependencies between the learned representation and the neighbor's representation. We calculate the neighbor representation matrix $z_{neigh} \in$

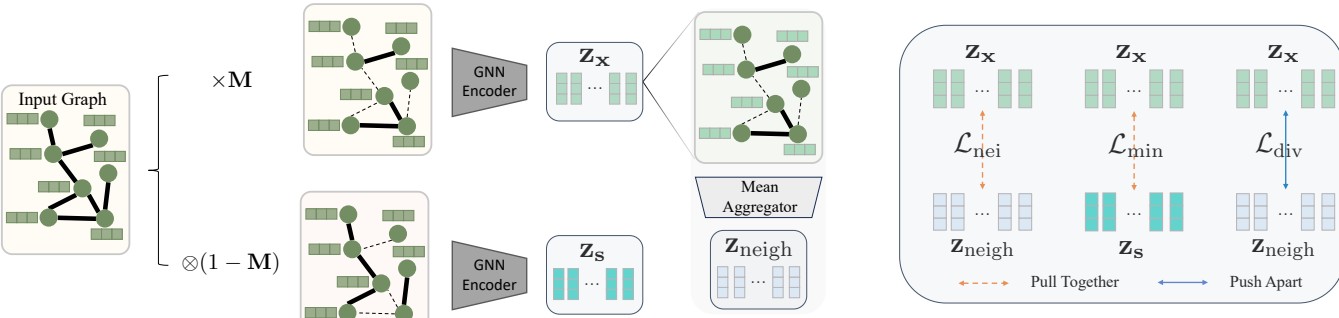

**Figure 2: Illustration of the BSG workflow. The left part shows the process of obtaining three representations $z_x$, $z_s$, and $z_{neigh}$. $z_s$ is the representation obtained by processing the complement of the masked graph $(1 - M)$. Furthermore, $z_{neigh}$ is acquired by applying a mean aggregator to $z_x$. The right part established three loss functions that additionally consider the neighbor information.**

$\mathbb{R}^{N \times D_{emb}}$ by taking the mean value of the neighbors' representations, where $N$ is the number of nodes and $D_{emb}$ is the dimension of the encoded representation, which can be easily obtained like:

$$\mathbf{z}_{\text{neigh}}[i, :] = \text{mean}(\{\mathbf{z}_{\mathbf{x}}[j, :] | j \in \mathcal{N}_i\}), \qquad (7)$$

where $\mathcal{N}_i$ indicates the neighbor set of node $i$. By assuming that $\mathbf{Z}_\mathbf{X}$ and $\mathbf{Z}_\mathbf{N}$ are zero-mean, unit-variance Gaussian vectors, $I(\mathbf{Z}_\mathbf{X}; \mathbf{Z}_\mathbf{N})$ can be expressed as:

$$I(\mathbf{Z}_\mathbf{X}; \mathbf{Z}_\mathbf{N}) = -\frac{D_{emb}}{2} \ln\left(1 - (1 - \frac{\text{MSE}}{2D_{emb}})^2\right) \approx -\frac{D_{emb}}{2} \ln(\frac{\text{MSE}}{D_{emb}}), \qquad (8)$$

where MSE is the mean squared distance between $\mathbf{z}_{\mathbf{x}}$ and $\mathbf{z}_{\text{neigh}}$. A detailed derivation of the equation is in the Appendix. From Equation (8), $I(\mathbf{Z}_\mathbf{X}; \mathbf{Z}_\mathbf{N})$ is explicitly increased when the MSE of two representations are minimized, leading to the neighbor loss expressed as:

$$\mathcal{L}_{\text{nei}} = \sum_{v_i \in \mathcal{V}} [\| \mathbf{z}_{\mathbf{x}}[v_i, :] - \mathbf{z}_{\text{neigh}}[v_i, :] \|_2^2]. \qquad (9)$$

Optimizing the neighbor loss can be interpreted in both SSL and graph perspectives. In the SSL context, objectives such as mask reconstruction and contrastive learning rely on assuming the multi-view assumption that the self-supervised supervision $\mathbf{S}$ does not change the original downstream task labels of $\mathbf{X}$ [30]. Under the homophily setting, where most of its neighbors are similar to each other, the neighbors of a node can be conceptualized as a distinct view of that node. From the graph perspective, akin to the aggregation mechanisms in Graph Neural Networks (GNNs), our neighbor loss is closely tied to the concept of graph smoothing.

### 4.4 Minimal Loss is Minimizing $H(\mathbf{Z}_\mathbf{X}|\mathbf{S})$

Minimizing $H(\mathbf{Z}_\mathbf{X}|\mathbf{S})$ encourages the reconstruction of the latent representation when the masked signal is given. Therefore, we can express it as maximizing $\mathbb{E}_{P_{\mathbf{S},\mathbf{Z}_\mathbf{X}}}[(\mathbf{Z}_\mathbf{X}|\mathbf{S})]$. According to Equation (6), BSG also maximizes $I(\mathbf{Z}_\mathbf{X}; \mathbf{S})$. Furthermore, the proposed minimal loss minimizes $H(\mathbf{Z}_\mathbf{X}|\mathbf{S})$, satisfying the condition of Equation (2) resembling the minimal representation. Therefore, by optimizing the minimal loss function, the representation extracts only the

task-relevant information. The conditional entropy $H(\mathbf{Z}_\mathbf{X}|\mathbf{S})$ can be expressed by assuming the probability distributions as Gaussian:

$$H(\mathbf{Z}_\mathbf{X}|\mathbf{S}) = \frac{1}{2} \log(2\pi e^{\text{var}(\mathbf{Z}_\mathbf{X}|\mathbf{S})}), \qquad (10)$$

where $\text{var}(\mathbf{Z}_\mathbf{X}|\mathbf{S})$ is the variance. Therefore, we can minimize $H(\mathbf{Z}_\mathbf{X}|\mathbf{S})$ as follows:

$$\mathcal{L}_{\text{min}} = \sum_{v_i \in \mathcal{V}} [\| \mathbf{z}_{\mathbf{x}}[v_i, :] - \mathbf{z}_{\mathbf{s}}[v_i, :] \|_2^2], \qquad (11)$$

where the masked representation $\mathbf{z}_{\mathbf{s}}$ can be easily obtained by processing the GNN model with the masked edges.

The downstream tasks for graphs, like node classification and link prediction, have different characteristics. Therefore, it is crucial to extract only the task-relevant information while discarding the task-irrelevant information. The minimal loss function facilitates the model in constructing representations that encapsulate meaningful, task-relevant information rather than simply replicating the input data. This approach enhances the model's ability to generalize effectively across diverse downstream tasks.

### 4.5 Divergence Loss is Maximizing $H(\mathbf{Z}_\mathbf{X}|\mathbf{Z}_\mathbf{N})$

There is a trade-off relation between the first term of Equation (6) $(I(\mathbf{Z}_\mathbf{X}; \mathbf{Z}_\mathbf{N}))$ and the last term $(H(\mathbf{Z}_\mathbf{X}|\mathbf{Z}_\mathbf{N}))$. Furthermore, maximizing the mutual information $I(\mathbf{Z}_\mathbf{X}; \mathbf{Z}_\mathbf{N})$ with the help of GNN and the proposed loss function $\mathcal{L}_{\text{nei}}$ may result in a degenerate solution, where $\mathbf{Z}_\mathbf{X}$ and $\mathbf{Z}_\mathbf{N}$ converge to identical representations. The mutual information maximizes when the two-term leads to an equal representation. In terms of self-supervised learning, when the representations of a node and its neighboring nodes are identical, indicating a high $I(\mathbf{Z}_\mathbf{X}; \mathbf{Z}_\mathbf{N})$, it becomes ineffective in distinguishing between the nodes which may underperform on some downstream tasks. In terms of graph data and GNN, this phenomenon is a well-known limitation, which refers to oversmoothing [13, 40]. Since oversmoothing is well known for leading to a drastic performance drop, we should penalize the two terms for being identical.

Therefore, we identify a node that has a similar representation to its neighbors and introduce a loss function designed to maximize the divergence between these representations. The similarity matrix

is obtained by taking the cosine similarity of the learned representation $\mathbf{z_x}$ and its neighbor representation $\mathbf{z}_{\text{neigh}}$. The similarity vector $\mathbf{R} \in \mathbb{R}^N$, where $N$ is the number of nodes in a graph, indicates the similarity between $\mathbf{Z_X}$ and $\mathbf{Z_N}$. The detail of the similarity vector and its following hinge loss is described as follows:

$$\mathbf{R} = \text{cosine\_similarity}(\mathbf{z_x}[i,:], \mathbf{z}_{\text{neigh}}[i,:]), \tag{12}$$

$$\mathcal{L}_{\text{div}} = \sum_{v_i \in \mathcal{V}} \max(0, \mathbf{R}[i] - m), \tag{13}$$

where $m$ is the margin for determining whether the node should be penalized.

## 4.6    SSL Objective is Maximizing $I(\mathbf{Z_X}; \mathbf{S})$

We utilize a simple edge mask reconstruction for $\mathbf{S}$, and we set 0.7 for the experiment. The masked adjacency matrix $\tilde{\mathbf{A}}$ can be defined as $\mathbf{M} \otimes \mathbf{A}$, where $\mathbf{M}$ is the edge mask and $\otimes$ is the Hadamard product. The representation $\mathbf{z_x}$ is obtained by processing the GNN-based encoder $f_E(\tilde{\mathbf{A}}, \mathbf{x})$. Then, the decoder model $f_D$ takes the learned representation as input to identify the masked edges. We adopt the MLP structure as a decoder. Similar to GAE [15], we additionally propose a mask reconstruction loss for identifying a simple graph structure. The loss function can be defined as:

$$\mathcal{L}_{\text{st}}^+ = \frac{1}{|\mathcal{E}^+|} \sum_{(u,v) \in \mathcal{E}^+} \log f_D(\mathbf{z}_{\mathbf{x}[\mathbf{u}]}, \mathbf{z}_{\mathbf{x}[\mathbf{v}]}),$$

$$\mathcal{L}_{\text{st}}^- = \frac{1}{|\mathcal{E}^-|} \sum_{(u',v') \in \mathcal{E}^-} \log(1 - f_D(\mathbf{z}_{\mathbf{x}[\mathbf{u}']}, \mathbf{z}_{\mathbf{x}[\mathbf{v}']})), \tag{14}$$

$$\mathcal{L}_{\text{st}} = -(\mathcal{L}_{\text{st}}^+ + \mathcal{L}_{\text{st}}^-),$$

where the positive edge set $\mathcal{E}^+$ is the set of masked edges and the negative edge set $\mathcal{E}^-$ is the randomly sampled edge set that is not included in the original edge set $\mathcal{E}$. $\mathcal{L}_{\text{st}}$ can be easily modified with different $\mathbf{S}$. For instance, the loss can be seamlessly replaced by a feature masking reconstruction loss when we adapt $\mathbf{S}$ as a feature mask, and the InfoNCE loss [23] when $\mathbf{S}$ indicates another positive view.

## 4.7    Training and Inference

Finally, the newly proposed loss functions of BSG can be summarized as:

$$\mathcal{L}_{\text{BSG}} = \lambda_1 \mathcal{L}_{\text{neigh}} + \lambda_2 \mathcal{L}_{\text{min}} + \lambda_3 \mathcal{L}_{\text{div}}, \tag{15}$$

where $\lambda_1$, $\lambda_2$, and $\lambda_3$ are the hyperparameters to adjust the effect of each term. BSG is optimized in conjunction with the graph structure loss function ($\mathcal{L}_{\text{st}}$), which in our case is reconstructing the masked edge. Therefore, the final loss function is defined as:

$$\mathcal{L} = \mathcal{L}_{\text{st}} + \mathcal{L}_{\text{BSG}}. \tag{16}$$

After learning with the loss function, $\mathbf{z_x}$ is utilized for the downstream tasks.

## 4.8    Analysis of BSG

### 4.8.1    Neighbor loss.
This section first shows that optimizing the neighbor loss function leads to graph smoothing where the representation $\mathbf{z_x}$ becomes similar with $\mathbf{z}_{\text{neigh}}$. Next, for the intricate inter-connected graph data, we demonstrate our representation has

higher mutual information than the existing sufficient SSL representation. We first define the graph embedding smoothness metric similar to [44]:

$$\delta = \frac{\| \sum_{v_i \in \mathcal{V}} (\sum_{v_j \in \mathcal{N}_i} (\mathbf{z}[v_i,:] - \mathbf{z}[v_j,:])^2) \|}{|\mathcal{E}| D_{emb}}. \tag{17}$$

With the smoothness metric, we can lead to the following theorem:

**THEOREM 1.** *For a graph $G$ and its feature vector $\mathbf{x}$, optimizing the neighbor loss is correlated to minimizing the graph embedding smoothness metric $\delta$, which attains the goal of graph smoothing.*

*proof sketch.* The theorem can be easily demonstrated since the neighbor representation $\mathbf{z}_{\text{neigh}}$ in the loss function can be exchanged as the empirical mean value of its neighbors. Therefore, it can be treated as a similar expression with the denominator of acquiring the graph embedding smoothness $\delta$. Similarly, GNN implicitly reduces the graph embedding smoothness metric, in which most existing graph-based SSL methodologies adopt a GNN encoder.

Furthermore, when the input graph is a homophilic graph indicating that the feature is similar to its neighbors, we can represent the following theorem as:

**THEOREM 2.** *Under the setting where the input is a homophilic graph, the representation obtained by optimizing $\mathcal{L}_{st}$ and $\mathcal{L}_{nei}$ satisfies the following relation:*

$$I(\mathbf{X}; \mathbf{Y}) \geq I(\mathbf{Z_X}^{BSG}; \mathbf{Y}) \geq I(\mathbf{Z_X}^{ssl}; \mathbf{Y}). \tag{18}$$

The theorem indicates that considering additional neighboring representation in inter-connected graph data leads to high mutual information with the downstream tasks. The detailed proof is provided in the Appendix. The neighbor loss allows our representation to be generated based solely on the input $\mathbf{X}$, even within the graph domain, whereas conventional methods require additional consideration of $\mathbf{N}$ to reflect the graph properties. Therefore, we can conclude that our representation has higher mutual information with the downstream task label than the original sufficient representation.

### 4.8.2    Minimal loss.
This section shows that optimizing the minimal loss together with the self-supervised learning loss leads to minimal representations. The relation of sufficient and minimal relation in the perspective of task-relevant is shown as:

$$I(\mathbf{Z_X}^{\text{SSL}}; \mathbf{X}|\mathbf{Y}) = I(\mathbf{X}; \mathbf{S}|\mathbf{Y}) + I(\mathbf{Z_X}^{\text{SSL}}; \mathbf{X}|\mathbf{S}, \mathbf{Y})$$
$$\geq I(\mathbf{Z_X}^{\text{SSL\_min}}; \mathbf{X}|\mathbf{Y}) = I(\mathbf{X}; \mathbf{S}|\mathbf{Y}). \tag{19}$$

The proof of the relation is provided in the Appendix. Similarly, we can lead to the following theorem that optimizing the minimal loss leads to generating the minimal representation as follows:

**THEOREM 3.** *Optimizing $\mathcal{L}_{min}$ and $\mathcal{L}_{st}$ leads to the representation $\mathbf{Z_X}^{BSG}$ which discards the task-irrelevant information similar to the minimal representation.*

$$I(\mathbf{Z_X}^{BSG}; \mathbf{X}|\mathbf{Y}) = I(\mathbf{Z_X}^{\text{SSL\_min}}; \mathbf{X}|\mathbf{Y}). \tag{20}$$

**Table 1: Node classification accuracy (%) on eight benchmark datasets. OOM denotes out-of-memory. In each column, the boldfaced score denotes the best result, and the underlined score represents the second-best result. A.R. indicates the average ranking across all datasets.**

| Model | Cora | Citeseer | Pubmed | Computers | Photo | CS | Physics | Arxiv | A.R. |
|-------|------|----------|--------|-----------|-------|-----|---------|-------|------|
| MLP | 50.98 ± 0.58 | 49.95 ± 0.26 | 68.40 ± 0.49 | 77.58 ± 0.82 | 85.84 ± 1.27 | 90.71 ± 0.05 | 94.45 ± 0.03 | 56.30 ± 0.30 | 16.00 |
| GCN | 81.50 ± 0.20 | 70.30 ± 0.40 | 79.00 ± 0.50 | 86.51 ± 0.54 | 92.42 ± 0.22 | 92.48 ± 0.21 | 95.38 ± 0.02 | 70.40 ± 0.30 | 9.38 |
| GAT | 83.00 ± 0.70 | 72.50 ± 0.70 | 79.00 ± 0.30 | 86.93 ± 0.29 | 92.56 ± 0.35 | 87.58 ± 2.86 | 94.19 ± 0.28 | 70.60 ± 0.30 | 10.00 |
| GAE | 76.25 ± 0.15 | 63.89 ± 0.18 | 77.24 ± 1.10 | 86.33 ± 0.44 | 91.71 ± 0.08 | 90.46 ± 0.29 | 93.04 ± 0.03 | 65.08 ± 0.24 | 15.38 |
| VGAE | 76.68 ± 0.17 | 64.34 ± 0.11 | 77.36 ± 0.60 | 86.70 ± 0.30 | 91.87 ± 0.04 | 92.33 ± 0.07 | 94.40 ± 0.07 | 67.70 ± 0.03 | 12.62 |
| ARGA | 77.95 ± 0.70 | 64.44 ± 1.19 | 80.44 ± 0.74 | 85.86 ± 0.11 | 91.82 ± 0.08 | 92.33 ± 0.07 | 94.32 ± 0.04 | 67.43 ± 0.08 | 12.75 |
| ARVGA | 79.50 ± 1.01 | 66.03 ± 0.65 | 81.51 ± 1.00 | 86.02 ± 0.11 | 91.51 ± 0.09 | 92.56 ± 0.09 | 93.64 ± 0.08 | 67.43 ± 0.08 | 11.75 |
| DGI | 81.44 ± 0.64 | 69.74 ± 2.48 | 78.83 ± 0.77 | 87.45 ± 0.47 | 91.65 ± 0.46 | 92.03 ± 0.05 | 94.89 ± 0.09 | 66.07 ± 0.45 | 12.25 |
| MVGRL | 82.22 ± 0.94 | 71.54 ± 0.85 | 79.46 ± 0.43 | 88.61 ± 0.64 | 92.64 ± 0.24 | 92.25 ± 0.03 | 95.10 ± 0.04 | 69.10 ± 0.10 | 8.50 |
| GRACE | 81.90 ± 0.40 | 71.20 ± 0.50 | 80.60 ± 0.40 | 86.25 ± 0.25 | 92.15 ± 0.24 | 91.90 ± 0.01 | 94.98 ± 0.05 | 68.70 ± 0.30 | 10.63 |
| CCA-SSG | 84.00 ± 0.40 | 73.10 ± 0.30 | 80.81 ± 0.38 | 88.76 ± 0.36 | 92.89 ± 0.28 | 93.01 ± 0.29 | 95.31 ± 0.07 | 69.22 ± 0.22 | 4.88 |
| MaskGAE | 83.58 ± 0.24 | 72.44 ± 0.17 | 82.00 ± 0.19 | 89.36 ± 0.18 | 92.79 ± 0.18 | 92.54 ± 0.21 | 95.15 ± 0.11 | 70.63 ± 0.30 | 5.25 |
| GraphMAE2 | 83.96 ± 0.85 | 73.42 ± 0.30 | 81.23 ± 0.57 | 87.42 ± 0.52 | 92.60 ± 0.11 | 91.31 ± 0.07 | 95.25 ± 0.05 | **71.77 ± 0.14** | 6.25 |
| GiGaMAE | 82.13 ± 0.80 | 70.04 ± 1.07 | 80.55 ± 0.75 | **90.20 ± 0.45** | 93.01 ± 0.46 | 92.54 ± 0.04 | 95.53 ± 0.03 | OOM | 6.14 |
| AUG-MAE | 84.10 ± 0.55 | 73.16 ± 0.44 | 81.12 ± 0.53 | 88.52 ± 0.17 | 92.82 ± 0.17 | 92.66 ± 0.19 | 95.62 ± 0.12 | 71.20 ± 0.30 | **4.00** |
| Bandana | 82.90 ± 0.39 | 71.39 ± 0.54 | 82.77 ± 0.49 | 89.28 ± 0.14 | 93.40 ± 0.10 | 92.79 ± 0.05 | 95.57 ± 0.03 | 71.04 ± 0.39 | 4.12 |
| BSG | **85.11 ± 0.26** | **74.63 ± 0.51** | **84.13 ± 0.23** | 89.99 ± 0.08 | **93.48 ± 0.16** | **93.19 ± 0.13** | **95.65 ± 0.03** | 71.25 ± 0.37 | **1.25** |

Given the relation in Equation (19) and if we can identify $Z_X$ with only the masked portion (S), which minimizes the conditional entropy $H(Z_X|S)$, it decreases $I(X; S|Y)$. Since the $\mathcal{L}_{st}$ increases $I(Z_X; S)$, it satisfies Equation (2). Therefore, the representation of BSG discards the task-irrelevant information, satisfying the minimal representation.

*4.8.3 Divergence loss.* This section shows that the divergence loss leads to alleviating the graph oversmoothing. We can easily identify that the divergence loss leads to an increase in the graph embedding smoothness metric $\delta$. To sum up, in the self-supervised domain in graphs, optimizing the neighbor loss and divergence loss is finding the balance between graph smoothing and oversmoothing. Since every dataset and each downstream task has its unique ideal graph smoothness metric, as mentioned in [13], it is crucial to consider both loss terms.

## 5 Experiment

We conduct several experiments to validate BSG. Specifically, we evaluate its performance on common downstream tasks such as node classification and link prediction, incorporating BSG into various SSL objectives, including node feature reconstruction and contrastive learning. Additionally, we assess its robustness to different mask ratios to determine whether BSG effectively extracts task-irrelevant information, and we analyze its sensitivity to hyperparameters. In the Appendix, we provide additional experiments, such as hyperparameter sensitivity to the margin introduced in $\mathcal{L}_{\text{div}}$, and the performance of BSG in graph classification tasks.

### 5.1 Experimental Setup

We conduct experiments on eight well-known benchmark datasets for the node classification, including citation network datasets (Cora, Citeseer, Pubmed) [26], Amazon co-purchase datasets (Computers, Photo) [27], coauthor dataset (CS, Physics) [27], and one

large scale dataset (Arxiv) [9]. The detailed experimental setting is provided in the Appendix.

Sixteen baselines are compared with BSG for the node classification, which can be categorized into four: basic semi-supervised learning models (MLP, GCN, GAT), generative and auto-encoding models (GAE [15], VGAE [15], ARGA [24], ARVGA [24]), contrastive learning models (DGI [35], MVGRL [49], GRACE [53], CCA-SSG [48]), and masking methods (MaskGAE [18], GraphMAE2 [7], GigaMAE [28], AUG-MAE [37], Bandana [51]). For the link prediction, we compare the self-supervised baselines only, a total of thirteen baselines.

### 5.2 Performance Comparison for Node Classification

We compare the accuracy of various methods on eight benchmark datasets, as shown in Table 1. Contrastive learning models slightly outperform generative models, and masking methods achieve higher performance. Recent baselines like AUG-MAE [37] and Bandana [51] achieve the second-best performance on most datasets, including Cora, Pubmed, Photo, and Physics datasets. However, their performance varies across different datasets, as seen in Bandana's results on the Citeseer dataset. In contrast, BSG consistently achieves high performance, often surpassing other models due to its ability to find a balance between the consideration of its neighbor information and the extraction of task-irrelevant information. Overall, BSG outperforms existing baselines in node classification tasks.

### 5.3 Performance Comparison for Link Prediction

For the link prediction task, we analyze BSG with twelve baselines on five benchmark datasets. For baselines that are not initially designed for link prediction, we train a linear classifier on the learned

**Table 2: Link prediction results (%) on five benchmark datasets. In each column, the boldfaced score denotes the best result, and the underlined score represents the second-best result. A.R. refers to the average ranking.**

| Model | Cora | | Citeseer | | Pubmed | | Computers | | Photo | | A.R |
|---|---|---|---|---|---|---|---|---|---|---|---|
| | AUC | AP | AUC | AP | AUC | AP | AUC | AP | AUC | AP | |
| GAE | 91.09 ± 0.26 | 92.83 ± 0.39 | 90.52 ± 0.45 | 91.68 ± 0.52 | 96.40 ± 1.23 | 96.50 ± 1.65 | 71.23 ± 1.03 | 68.77 ± 0.80 | 71.45 ± 1.01 | 66.04 ± 0.96 | 11.0 |
| VGAE | 91.40 ± 0.01 | 92.60 ± 0.01 | 90.80 ± 0.02 | 92.00 ± 0.02 | 94.40 ± 0.02 | 94.70 ± 0.02 | 92.69 ± 0.03 | 88.27 ± 0.08 | 95.61 ± 0.05 | 94.63 ± 0.06 | 8.3 |
| ARGA | 92.40 ± 0.18 | 93.23 ± 0.20 | 91.94 ± 0.35 | 93.03 ± 0.25 | 96.81 ± 0.06 | 97.11 ± 0.06 | 67.28 ± 2.91 | 62.83 ± 2.63 | 85.43 ± 0.82 | 81.58 ± 1.40 | 9.3 |
| ARVGA | 92.40 ± 0.89 | 92.60 ± 0.85 | 92.40 ± 0.33 | 93.00 ± 0.33 | 96.50 ± 0.32 | 96.80 ± 0.41 | 92.38 ± 0.15 | 88.49 ± 0.33 | 95.44 ± 0.14 | 94.51 ± 0.12 | 7.0 |
| DGI | 93.88 ± 1.00 | 93.60 ± 1.14 | 95.98 ± 0.72 | 96.18 ± 0.68 | 96.30 ± 0.20 | 95.65 ± 0.26 | 91.34 ± 1.23 | 91.13 ± 1.00 | 91.39 ± 1.42 | 90.63 ± 1.29 | 7.0 |
| MVGRL | 93.33 ± 0.68 | 92.95 ± 0.82 | 88.66 ± 5.27 | 89.37 ± 4.55 | 95.89 ± 0.22 | 95.53 ± 0.30 | 91.48 ± 2.09 | 91.07 ± 1.89 | 91.72 ± 0.88 | 90.94 ± 0.86 | 9.3 |
| GRACE | 82.67 ± 0.27 | 82.36 ± 0.24 | 87.74 ± 0.96 | 86.92 ± 1.11 | 94.09 ± 0.92 | 93.26 ± 1.20 | 89.97 ± 0.25 | 92.15 ± 0.43 | 88.64 ± 1.17 | 83.85 ± 2.63 | 12.4 |
| CCA-SSG | 93.88 ± 0.95 | 93.74 ± 1.15 | 89.53 ± 0.95 | 90.13 ± 0.91 | 94.09 ± 0.45 | 93.52 ± 0.35 | 83.85 ± 1.35 | 84.04 ± 1.74 | 91.04 ± 1.98 | 89.68 ± 2.05 | 10.3 |
| MaskGAE | 96.42 ± 0.27 | 96.05 ± 0.16 | 97.74 ± 0.14 | 97.99 ± 0.12 | 98.74 ± 0.09 | 98.64 ± 0.06 | 98.56 ± 0.02 | 98.38 ± 0.03 | 98.49 ± 0.05 | 98.26 ± 0.07 | 2.0 |
| GraphMAE2 | 94.88 ± 0.23 | 93.52 ± 0.51 | 94.35 ± 0.45 | 95.25 ± 0.41 | 96.71 ± 0.12 | 96.32 ± 0.11 | 91.62 ± 0.43 | 89.69 ± 0.56 | 93.34 ± 0.40 | 91.33 ± 0.48 | 6.4 |
| GigaMAE | 94.48 ± 0.12 | 94.09 ± 0.21 | 95.11 ± 0.11 | 95.41 ± 0.11 | 93.56 ± 0.82 | 92.42 ± 0.92 | 94.01 ± 0.20 | 91.21 ± 0.35 | 95.01 ± 0.67 | 93.04 ± 0.60 | 6.7 |
| AUG-MAE | 90.51 ± 0.57 | 89.82 ± 0.62 | 90.63 ± 0.79 | 91.44 ± 0.90 | 94.69 ± 0.71 | 94.10 ± 0.88 | 87.40 ± 0.21 | 86.62 ± 0.34 | 92.25 ± 0.84 | 91.43 ± 0.96 | 10.3 |
| Bandana | 95.83 ± 0.06 | 95.38 ± 0.11 | 96.70 ± 0.31 | 97.04 ± 0.38 | 97.31 ± 0.09 | 96.82 ± 0.24 | 97.27 ± 0.14 | 96.91 ± 0.22 | 97.33 ± 0.08 | 96.92 ± 0.12 | 3.1 |
| BSG | **96.69 ± 0.08** | **96.34 ± 0.13** | **98.14 ± 0.10** | **98.34 ± 0.09** | **98.91 ± 0.02** | **98.87 ± 0.05** | **98.60 ± 0.02** | **98.40 ± 0.03** | **98.70 ± 0.02** | **98.50 ± 0.03** | **1.0** |

**Table 3: Link prediction results (%) on five benchmark datasets. We compared the strongest baseline MaskGAE with the dot product method.**

| Dataset | MaskGAE | | BSG | |
|---|---|---|---|---|
| | AUC | AP | AUC | AP |
| Cora | 95.19 ± 0.24 | 94.95 ± 0.24 | **95.92 ± 0.13** | **95.53 ± 0.14** |
| Citeseer | 97.28 ± 0.18 | 97.64 ± 0.19 | **97.66 ± 0.25** | **97.84 ± 0.19** |
| Pubmed | 95.42 ± 0.69 | 96.45 ± 0.46 | **97.66 ± 0.58** | **97.51 ± 0.08** |
| Computers | 91.48 ± 1.54 | 86.45 ± 2.74 | **94.29 ± 0.58** | **88.70 ± 1.39** |
| Photo | 91.71 ± 0.61 | 86.06 ± 0.93 | **93.25 ± 0.81** | **88.70 ± 1.39** |

representations, similar to the setting in [18]. We evaluate using AUC and AP metrics, and BSG achieves the highest performance across all datasets, as shown in Table 2. Additionally, baselines that perform well in the node classification task tend to show lower performance in the link prediction task. This suggests that without taking neighbor information into account, they may fail to generalize as effectively across different tasks, while BSG remains consistent on different downstream tasks by balancing the representations in terms of graph embedding smoothness. Furthermore, since both MaskGAE and BSG use the similar edge decoder that we employed in optimizing $\mathcal{L}_{st}$, we extend our comparison to include the MaskGAE model, evaluating link prediction performance using a simple dot product method. The results, shown in Table 3, indicate that BSG performs better than MaskGAE under this evaluation approach, especially in the Computers and Photo dataset. From the result, we can conclude that considering its neighbor information for finding the balance in terms of graph smoothness is more effective in identifying the underlying graph properties and achieves outperformance in link prediction tasks.

## 5.4 Analysis

In this section, we conduct additional experiments over BSG to gain a deeper understanding of their individual contributions. We validate the effect of $\mathcal{L}_{nei}$ and $\mathcal{L}_{div}$ in terms of graph smoothness whether each loss empirically relates to smoothing or oversmoothing. We apply BSG to different graph SSL objectives, including contrastive learning and node feature reconstruction. Moreover, we perform the hyperparameter sensitivity with the existence of loss

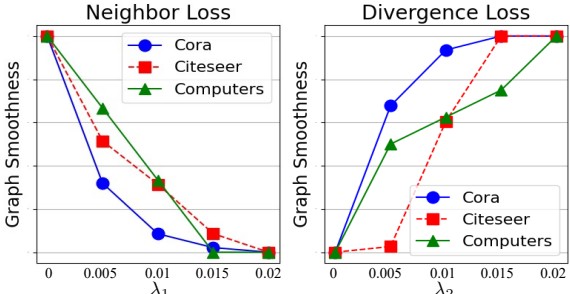

**Figure 3: The effect of $\mathcal{L}_{nei}$ and $\mathcal{L}_{div}$ respect to graph smoothness. The y-axis denotes the normalized graph embedding smoothness score, and the low values indicate oversmoothing.**

functions, the effect of mask-ratio, and the margin that is utilized in the divergence loss.

*5.4.1 Empirical Analysis of $\mathcal{L}_{nei}$ and $\mathcal{L}_{div}$.* We conduct a comprehensive experiment to assess the effects of two trade-off-related loss functions: neighbor loss and divergence loss. The neighbor loss promotes graph smoothing, while the divergence loss counteracts this effect, preventing the representation from becoming oversmoothed. To quantify the impact of each loss function, we evaluate their performance using graph embedding feature smoothness, where a lower score indicates greater oversmoothing. Moreover, when identifying the effect of each loss, we do not use different loss functions to fully analyze the corresponding loss. As illustrated in Figure 3, the neighbor loss reduces the feature smoothness score, while the divergence loss increases it. By adjusting these two loss functions, BSG achieves a more balanced representation, effectively capturing the underlying structure of the input graph. This trend is consistently observed across different datasets, as depicted in the figure.

*5.4.2 Incorporating BSG into Other SSL Objectives.* Table 4 elucidates the effect of BSG by extending the loss function we propose in other SSL objectives, such as contrastive learning and node feature reconstruction. For simplicity, we modify $\mathcal{L}_{st}$ into [48] for

**Table 4: Downstream task accuracy (%) when incorporating BSG into other graph SSL baselines for the node classification task. CL: Contrastive Learning, FR: Feature Reconstruction, and ER stands for Edge Reconstruction.**

| Dataset | Metric | CL | CL + BSG | FR | FR + BSG | ER | ER + BSG |
|---------|--------|-------|----------|-------|----------|-------|----------|
| Cora | ACC | 84.00 | 84.22 | 83.96 | 84.68 | 83.33 | 85.11 |
| | AUC | 93.88 | 94.70 | 94.88 | 96.66 | 96.42 | 96.69 |
| | AP | 93.74 | 94.64 | 93.52 | 96.64 | 95.91 | 96.34 |
| Citeseer | ACC | 73.10 | 73.36 | 73.42 | 73.60 | 73.02 | 74.63 |
| | AUC | 89.53 | 94.41 | 94.35 | 96.07 | 97.96 | 98.14 |
| | AP | 90.13 | 94.68 | 95.25 | 95.40 | 98.12 | 98.34 |
| Pubmed | ACC | 80.81 | 81.26 | 81.23 | 81.65 | 81.77 | 84.13 |
| | AUC | 94.09 | 96.61 | 96.71 | 97.67 | 98.55 | 98.91 |
| | AP | 93.52 | 96.06 | 96.32 | 97.53 | 98.32 | 98.87 |

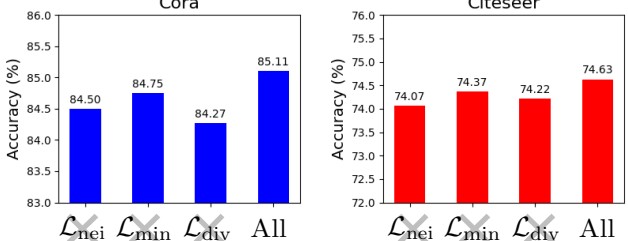

**Figure 4: Ablation study of the proposed loss functions. We compared the performance without applying each loss function.**

contrastive learning and [8] for node feature reconstruction. The table indicates that original contrastive learning and feature reconstruction methods relatively outperform node classification over edge reconstruction based methods. By applying BSG, the representation achieves a more balanced state in terms of graph smoothness, resulting in a significant improvement in the link prediction task and a slight improvement in the node classification task. On the other hand, when applying edge reconstruction as $\mathcal{L}_{st}$, link prediction relatively outperforms the node classification task. By incorporating BSG, the node classification performance surpasses the performance compared to the original contrastive and feature reconstruction methods. The extended table with the standard deviation is provided in the Appendix.

*5.4.3 Ablation of Loss Functions.* Figure 4 illustrates the ablation study on the effects of each loss function by comparing performance when each function is omitted. The results indicate that utilizing all proposed loss functions yields the best performance. Notably, divergence loss plays a crucial role in enhancing performance on the Cora dataset, while neighbor loss proves to be more impactful on the Citeseer dataset. This suggests that different datasets possess distinct features, highlighting the necessity of smoothing in some cases and the importance of preventing oversmoothing in others. Unlike other existing studies that do not address these two terms directly, BSG stands out as the only self-supervised learning model capable of achieving a balance between smoothing and oversmoothing.

*5.4.4 Mask Ratio.* Existing studies that utilize edge reconstruction methods typically employ a high mask ratio, such as 0.7. However,

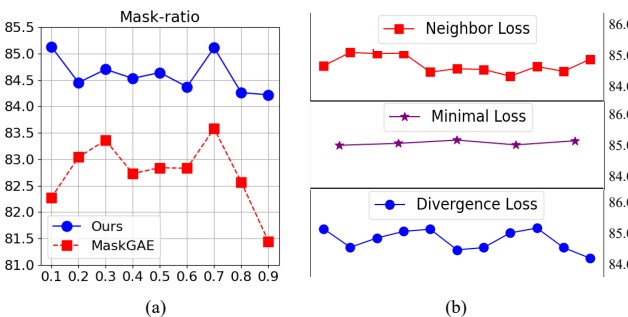

**Figure 5: Extended experiment of BSG in the Cora dataset, with values indicating the node classification results. (a) compares the performance of BSG and MaskGAE with different mask ratios. (b) analyzes the sensitivity of loss functions.**

a low mask ratio tends to overemphasize the proximity of existing edges, which may not correlate with downstream node classification tasks, while a high mask ratio complicates the identification of the original graph structure due to the substantial number of masked edges. In contrast, minimal loss in BSG effectively extracts task-irrelevant information from the edge mask, enhancing its relevance to the downstream task. As illustrated in Figure 5-(a), BSG demonstrates increased robustness across varying edge mask ratios.

*5.4.5 Sensitivity of Loss Functions.* Figure 5-(b) illustrates the sensitivity of each loss function. The search space for every loss function is denoted in the Appendix. The figure demonstrates a slight inverse correlation between the performance of the neighbor loss and divergence loss, which aligns with the inherent trade-off between these objectives. Despite the potential complex optimization due to the numerous loss functions, BSG demonstrates robustness to hyperparameter variations, underscoring its effectiveness and reliability.

## 6 Conclusion

We propose BSG to consider the underlying graph features in terms of graph smoothness, which is underexplored in graph-based SSL. Our approach is centered on adding an additional variable, the neighbor variable, through the Information Theory. This transformation introduces three terms, and each term is controlled with each loss function. The neighbor loss function maximizes the mutual information with the node and its neighbors, and the divergence loss has a trade-off relation to the neighbor loss, which prevents the representation from falling into oversmoothing. The minimal loss is additionally proposed to discard the task-irrelevant information, and satisfies the minimal representation in SSL. We theoretically demonstrate that each loss function has its functionality in terms of both the SSL perspective and graph perspective. Experimental results on graph-related downstream tasks and further experiments consistently show that BSG significantly outperforms recent baselines on real-world datasets. These findings highlight the effectiveness of our approach in enhancing the quality of graph representations and establishing BSG as a robust solution for self-supervised learning in graphs.

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

**Table 5: Data statistics for datasets used in node classification.**

|  | # of nodes | # of edges | Feature dimension | # of classes | Edge density |
|---|---|---|---|---|---|
| Cora | 2,708 | 10,556 | 1,433 | 7 | 1.44 |
| Citeseer | 3,327 | 9,104 | 3,703 | 6 | 0.82 |
| Pubmed | 19,717 | 88,648 | 500 | 3 | 0.23 |
| Computers | 13,381 | 245,778 | 767 | 10 | 2.60 |
| Photo | 7,487 | 119,043 | 745 | 8 | 4.07 |
| CS | 18,333 | 81,894 | 6,805 | 15 | 0.24 |
| Physics | 34,493 | 247,962 | 8,415 | 5 | 0.21 |
| Arxiv | 169,343 | 2,315,598 | 128 | 40 | 0.08 |

# A Appendix

## A.1 Proof of Equations and Theorems

A.1.1 *Equation* (8). Equation (8) in the main text restated as:

$$I(Z_X, Z_N) = -\frac{D_{emb}}{2}\ln(1 - (1 - \frac{MSE}{2D_{emb}})^2) \approx -\frac{D_{emb}}{2}\ln(\frac{MSE}{D_{emb}}). \tag{21}$$

There is an exact relationship between the $I$ and the correlation coefficient $\rho$ as follows where assuming $Z_X$ and $Z_N$ as bivariate normal distribution:

$$I(Z_X; Z_N) = -\frac{D_{emb}}{2}\ln(1 - \rho^2). \tag{22}$$

Similarly, the mean squared error (MSE) loss between $Z_X$ and $Z_N$ is represented as:

$$MSE = \mathbb{E}[||Z_X - Z_N||^2] = 2D_{emb}(1 - \rho), \tag{23}$$

where we assume both variables are zero-mean and unit-variance Gaussian random variables. From Equation (23):

$$MSE = 2D_{emb}(1 - \rho)$$
$$\rho = 1 - \frac{MSE}{2D_{emb}}. \tag{24}$$

Substituting $\rho$ into Equation (22) results in:

$$I(Z_X; Z_N) = -\frac{D_{emb}}{2}\ln\left(1 - \left(1 - \frac{MSE}{2D_{emb}}\right)^2\right)$$
$$= -\frac{D_{emb}}{2}\ln\left(\frac{MSE}{D_{emb}} - \frac{MSE^2}{4D_{emb}^2}\right). \tag{25}$$

Assuming MSE is small compared to $D_{emb}$, we can neglect the second-order term:

$$1 - \left(1 - \frac{MSE}{2D_{emb}}\right)^2 \approx \frac{MSE}{D_{emb}}. \tag{26}$$

Therefore, we can derive Equation (8).

A.1.2 *Theorem 2 (Neighbor Loss)*. Restate Theorem 2:

**Theorem 2.** *Under the setting where the input is a homophilic graph, the representation obtained by optimizing $\mathcal{L}_{st}$ and $\mathcal{L}_{nei}$ satisfies the following relation:*

$$I(X; Y) \geq I(Z_X^{BSG}; Y) \geq I(Z_X^{ssl}; Y). \tag{27}$$

(See Theorem 2.)

Proof. The proof is contained with two parts. We first show that $I(X; Y) \geq I(Z_X^{ssl}; Y)$ in all domains including graph indicating that the upperbound of $I(Z_X^{ssl}; Y)$ is $I(X; Y)$. Then we additionally show that in the graph domain, BSG that considers the neighbor

information has a higher mutual information than the general sufficient representation of SSL.

$I(X; Y) \geq I(Z_X^{ssl}; Y)$: By adding a new variable $Y$ in $I(X; S)$ and $I(Z_X; S)$, we can express like:

$$I(X; S) = I(X; Y; S) + I(X; S|Y)$$
$$= I(X; Y) - I(X; Y|S) + I(X; S|Y).$$
$$I(Z_X; S) = I(Z_X; Y; S) + I(Z_X; S|Y)$$
$$= I(Z_X; Y) - I(Z_X; Y|S) + I(Z_X; S|Y).$$

For the general self-supervised setting, the Markov Chain $S \leftrightarrow Y \leftrightarrow X \to Z_X$ holds. From the Markov Chain, we can easily acknowledge that the maximum value of $I(Z_X; S)$ is $I(X; S)$. Therefore, we can exchange the variable $X$ as $Z_X^{ssl}$. Then, the following equation holds:

$$I(Z_X^{ssl}; S) = I(Z_X^{ssl}; Y) - I(Z_X^{ssl}; Y|S) + I(Z_X^{ssl}; S|Y)$$
$$= I(X; Y) - I(X; Y|S) + I(Z_X^{ssl}; S|Y).$$

Then we can easily demonstrate that $I(X; Y) \geq I(Z_X^{ssl}; Y)$ since $I(X; Y|S) \geq I(Z_X^{ssl}; Y|S)$.

$I(Z_X^{BSG}; Y) \geq I(Z_X^{ssl}; Y)$: For the self-supervised setting with graphs, the Markov Chain $S \leftrightarrow Y \leftrightarrow X \to Z_X$, and $N \to Z_X$ holds. The Markov Chain we utilizes in the first part neglects the extra relation $N \to Z_X$. Therefore, unlike different domains that do not need consideration of additional variable $N$, for the self-supervised setting in graphs, there is an extra space between $I(Z_X^{ssl}; Y)$ and $I(X; Y)$. We first begin by introducing a new Lemma, which can simplify the Markov Chain.

**Lemma 1.** *If $P(Z_X^{BSG}|X)$ is Dirac, then we can induce the Markov chain as $S \leftrightarrow Y \leftrightarrow X \to Z_X^{BSG}$.*

From the lemma and an additional relation $N \to Z_X$, we can conclude that if $Z_X^{BSG}$ is solely deterministic with $X$ indicating that $N$ does not give any additional information when $X$ is given, we can conclude that in graphs, $I(Z_X^{BSG}; Y) \geq I(Z_X^{ssl}; Y)$ holds. Since $X$ and $N$ is related to the input graph, $H(X, Y)$ is constant. From the equation expressed as:

$$H(X, Y) = H(X|N) + H(N|X) - I(X; N)$$
$$\approx H(Z_X|Z_N) + H(Z_N|Z_X) - I(Z_X; Z_N),$$

we can lead to the following:

- when $I(Z_X; Z_N)$ is maximized, the conditional entropy $H(X|N)$ and $H(N|X)$ is minimized, denoting that the variables contains subtle information when the other one is given.
- $\mathcal{L}_{nei}$ explicitly maximizes $I(Z_X; Z_N)$.

From the two facts, even in the graph domain, the additional relation $N \to Z_X$ can be neglected and therefore, we can conclude that $I(Z_X^{BSG}; Y)$ is the intrinsic sufficient representation in the graph domain, which satisfies $I(Z_X^{BSG}; Y) \geq I(Z_X^{ssl}; Y)$, and we conclude the proof $\qquad\square$

A.1.3 *Theorem 3 (Minimal Loss)*. Restate Theorem 3.

**Theorem 3.** *Optimizing $\mathcal{L}_{min}$ and $\mathcal{L}_{st}$ leads to the representation $Z_X^{BSG}$ which discards the task-irrelevant information similar to the minimal representation.*

$$I(Z_X^{BSG}; X|Y) = I(Z_X^{SSL\_min}; X|Y). \tag{28}$$

**Table 6: Hyperparameters of BSG on node classification tasks.**

| dataset | $\lambda_1$ | $\lambda_2$ | $\lambda_3$ | $m$ |
|---------|-------------|-------------|-------------|-----|
| Cora | 0.0002 | 0.001 | 0.0009 | -0.2 |
| Citeseer | 0.0007 | 0.0001 | 0.0006 | 0.1 |
| Pubmed | 0.001 | 1e-05 | 0.0001 | -0.4 |
| Computers | 0.0006 | 1e-05 | 0.001 | -0.3 |
| Photo | 0.0001 | 1e-05 | 0.0005 | 0.5 |
| CS | 0.0004 | 0.1 | 0.0001 | 0.0 |
| Physics | 0.0009 | 0.0001 | 0.0007 | 0.1 |
| Arxiv | 0.0001 | 0.0001 | 0.01 | 0.2 |

(See Theorem 2.)

PROOF. We first begin by demonstrating Equation (19). Let's first recall the equation:

$$I(\mathbf{Z}_X^{SSL}; \mathbf{X}|\mathbf{Y}) = I(\mathbf{X}; \mathbf{S}|\mathbf{Y}) + I(\mathbf{Z}_X^{SSL}; \mathbf{X}|\mathbf{S}, \mathbf{Y})$$
$$\geq I(\mathbf{Z}_X^{SSL\_min}; \mathbf{X}|\mathbf{Y}) = I(\mathbf{X}; \mathbf{S}|\mathbf{Y}). \tag{29}$$

From the proof of the previous theorem and lemma, BSG can simplify the Markov Chain. Also, since the objective of SSL is to maximize $I(\mathbf{Z}_X; \mathbf{S})$, $I(\mathbf{Z}_X^{SSL}; \mathbf{S}|\mathbf{Y}) = I(\mathbf{Z}_X^{SSL\_min}; \mathbf{S}|\mathbf{Y}) = I(\mathbf{Z}_X^{BSG}; \mathbf{S}|\mathbf{Y}) = I(\mathbf{X}; \mathbf{S}|\mathbf{Y})$ holds. With the two conditions and the fact that $H(\mathbf{Z}_X|\mathbf{S})$ is minimized, $I(\mathbf{Z}_X^{SSL\_min}; \mathbf{X}|\mathbf{S}; \mathbf{Y}) = 0$, and we can conclude the proof. Furthermore, since $H(\mathbf{Z}_X|\mathbf{S})$ is minimized with the minimal loss, BSG also satisfies the minimal condition, demonstrating extracting task-irrelevant information. □

## A.2 Data Statistics

Table 5 present the data statistics we utilized on node classification and link prediction. The hyperparameters for the node classification are denoted in Table 6. For the link prediction, we tested with the default hyperparameters.

## A.3 Experimental Setting

For the dataset, we utilize the standard data split provided by the Pytorch geometric library or 1:1:8 (train:valid:test) for the dataset that is not provided for the node classification task. We run the experiments 10 times by fixing the seed for a fair comparison. The node classification performance is validated by comparing the mean accuracy and the standard deviation. We also evaluate the link prediction tasks with similar settings with node classification task. We conducted AUC and AP to compare the performance of the link prediction. The baseline performance is conducted by following the hyperparameters suggested by the original paper. Every experiment was held on a single NVIDIA-A100 GPU. We provide an anonymous GitHub for BSG with the following URL: https://anonymous.4open. science/r/BSG-2025/. The hyperparameter of controlling the loss functions ($\lambda_1$, $\lambda_2$, and $\lambda_3$) is tuned using a grid search: $\lambda_1$ and $\lambda_3$ is tuned by varying a range of 0.0001 to 0.001. $\lambda_2$ is chosen in [0.1, 0.01, 0.001, 0.0001, 0.00001].

## A.4 Extended Experiments

*A.4.1 Margin in Divergence Loss.* We introduced an additional margin, denoted as $m$, when calculating $\mathcal{L}_{div}$. Figure 6 illustrates

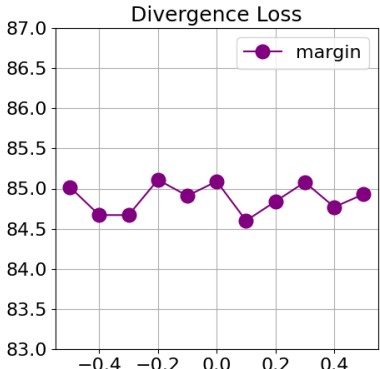

**Figure 6: Extended experiment of BSG on the Cora dataset, analyzing the sensitivity of three loss functions, with values representing the node classification performance (%).**

**Table 7: Extended version of Table 4 by adding the standard deviation.**

| Dataset | Metric | CL | BSG + CL | FR | BSG + FR | ER | BSG + ER |
|---------|--------|-----|----------|-----|----------|-----|----------|
| | ACC | 84.00 ± 0.40 | 84.22 ± 0.38 | 83.96 ± 0.85 | 84.68 ± 0.54 | 83.33 ± 0.25 | 85.11 ± 0.26 |
| Cora | AUC | 93.88 ± 0.95 | 94.70 ± 0.52 | 94.88 ± 0.23 | 96.66 ± 0.20 | 96.42 ± 0.17 | 96.69 ± 0.08 |
| | AP | 93.74 ± 1.15 | 94.64 ± 0.49 | 93.52 ± 0.51 | 96.64 ± 0.27 | 95.91 ± 0.25 | 96.34 ± 0.13 |
| | ACC | 73.10 ± 0.30 | 73.36 ± 0.52 | 73.42 ± 0.30 | 73.60 ± 0.35 | 73.02 ± 0.30 | 74.63 ± 0.51 |
| Citeseer | AUC | 89.53 ± 0.95 | 94.41 ± 0.04 | 94.35 ± 0.45 | 96.07 ± 0.69 | 97.96 ± 0.22 | 98.14 ± 0.10 |
| | AP | 90.13 ± 0.91 | 94.68 ± 0.04 | 95.25 ± 0.41 | 95.40 ± 0.68 | 98.12 ± 0.21 | 98.34 ± 0.09 |
| | ACC | 80.81 ± 0.38 | 81.26 ± 0.46 | 81.23 ± 0.57 | 81.65 ± 0.47 | 81.77 ± 0.57 | 84.13 ± 0.23 |
| Pubmed | AUC | 94.09 ± 0.45 | 96.61 ± 0.05 | 96.71 ± 0.12 | 97.67 ± 0.29 | 98.55 ± 0.04 | 98.91 ± 0.02 |
| | AP | 93.52 ± 0.35 | 96.06 ± 0.06 | 96.32 ± 0.11 | 97.53 ± 0.63 | 98.32 ± 0.06 | 98.87 ± 0.05 |

the sensitivity of the model to different margin values. As shown in the figure, BSG consistently performs well, regardless of the margin value. The goal of the divergence loss is to adjust the weight so that the model does not generate identical representations to those of neighboring nodes. Since identical representations have a cosine similarity of 1, a margin ranging from -0.5 to 0.5 effectively identifies representations that closely resemble those of neighbors, thereby ensuring consistent performance.

*A.4.2 Extended Tables.* We provide the details of Table 4 by including the standard deviation. Table 7 in the appendix shows the standard deviation. From the table, incorporating BSG is more robust on most datasets and settings.

*A.4.3 Graph Classification Task.* We established an extended experiment on graph classification tasks. Since node feature reconstruction is a representative method in Self-supervised learning (SSL) for graph classification, we apply the node feature reconstruction as $\mathcal{L}_{st}$. Table 8 shows the performance of BSG in the graph classification performance. BSG in graph classification task is compared with supervised models (GIN [42], DiffPool [45]), graph kernel based models (WL [38], DGK [43]), and self supervised models. (Graph2vec [22], infograph [31], GraphCL [47], JOAO [46], MVGRL [49], InfoGCL [41], GraphMAE [8]). The statistics of the dataset evaluated for the graph classification are shown in Table 9. While the supervised GIN [42] model outperformed most of the self-supervised model, BSG achieved state-of-the-art performance on all three datasets. The graph classification is the graph-level

**Table 8: Graph classification results (%) on three public benchmark datasets. In each column, the boldfaced score denotes the best result, and the underlined score represents the second-best result.**

| Model | | PROTEINS | IMDB-B | IMDB-M |
|---|---|---|---|---|
| Supervised | GIN | 76.20 ± 2.8 | 75.10 ± 5.1 | 52.30 ± 2.8 |
| | DiffPool | 75.10 ± 3.5 | 72.60 ± 3.9 | 50.30 ± 3.6 |
| Graph kernels | WL | 72.90 ± 0.6 | 72.30 ± 3.4 | 47.00 ± 0.5 |
| | DGK | 73.30 ± 0.8 | 67.00 ± 0.6 | 44.60 ± 0.5 |
| Self supervised | Graph2vec | 73.30 ± 2.1 | 71.10 ± 0.5 | 50.40 ± 0.9 |
| | infograph | 74.40 ± 0.5 | 73.00 ± 0.9 | 49.70 ± 0.5 |
| | GraphCL | 74.40 ± 0.5 | 71.10 ± 0.4 | 48.60 ± 0.7 |
| | JOAO | 74.60 ± 0.4 | 70.20 ± 3.1 | 49.20 ± 0.8 |
| | MVGRL | 75.00 ± 2.9 | 74.20 ± 0.7 | 51.20 ± 0.5 |
| | InfoGCL | 72.96 ± 3.0 | 75.10 ± 0.9 | 51.40 ± 0.8 |
| | GraphMAE | 75.30 ± 0.5 | 75.52 ± 0.3 | 51.62 ± 0.9 |
| | AUG-MAE | 75.49 ± 0.4 | 75.12 ± 0.5 | **53.07 ± 2.6** |
| | BSG | **76.21 ± 0.4** | **75.74 ± 0.6** | 52.48 ± 0.5 |

**Table 9: Data statistics on dataset applied to perform graph classification.**

| | # of graphs | # of classes | Avg. # of nodes |
|---|---|---|---|
| PROTEINS | 1,113 | 2 | 39.1 |
| IMDB-B | 1,000 | 2 | 19.8 |
| IMDB-M | 1,500 | 3 | 13.0 |

downstream task, which is advantageous if the model has high representation power. Therefore, it is critical to identify the underlying graph features, including the graph smoothness. BSG, which considers the graph smoothness by the proposed three loss functions, effectively captures the graph representation that leads to the outperformance even in the graph classification tasks.