# OpenReview forum: "Balancing Graph Embedding Smoothness in Self-supervised Learning via Information-Theoretic Decomposition"
_ACM.org/TheWebConf/2025/Conference — WWW 2025 Poster_

### Official Review · Reviewer_1G3M · 2024-11-28

**Novelty:** 4
**Technical Quality:** 5

**Review:**

Summary:

The authors propose the BSG framework, which aims to balance the smoothness of the learned embeddings through the design of a mutual information-based self-supervised loss function, thereby improving the model's performance on downstream tasks.

Strengths:

1.	The proposed method is interesting and straightforward, which makes it easy to understand and apply.
2.	The experiments are comprehensive, covering both link prediction and node classification tasks, which provides a well-rounded evaluation of the proposed method.

Weaknesses:

1.	The authors should give a thoroughly explain and discuss the two terms (first and last) in Equation (6) that represent a trade-off, as well as clarify why directly optimizing these terms leads to the desired balance in embedding smoothness, The discussion in Section 4.5 does not fully address my concerns. Further clarification is needed to resolve the remaining uncertainties.
2.	If balancing embedding smoothness is key to the improvement of SSL performance, the authors should calculate and report the specific embedding smoothness values for both their proposed method and all the baseline methods used for comparison. This would further strengthen the authors' argument. Figure 1 appears to be more of a case study, and additional quantitative analysis would help support the claims.
3.	The ablation study results in Figure 4 show very small differences, which are not statistically significant.
4.	There appear to be some typos in the paper, such as the bold formatting in the last column of Table 1, where two values are bolded.

**Questions:**

1.	Based on the conditional entropy computing formula of mutual information, the addition of the random variable $Z_N$ seems to directly lead to Equation (5), without the need for the transformations in Equations (3) and (4).
2.	If the ultimate goal is to balance embedding smoothness, why not directly add an embedding smoothness regularization term to the existing SSL loss, controlled by a hyperparameter to adjust the loss weight? This could achieve the same goal of balancing embedding smoothness. What advantages does using mutual information offer in this context? Based on the experimental results, it appears that the current balance is also achieved through the selection of hyperparameters.
3.	I am unable to interpret Figure 5(b). Could the authors clarify what the x-axis represents?
4.	What is the experimental setting (e.g., dataset split, number of repetitions) for the graph classification task in Appendix A.4?

**Reviewer Confidence:**

3: The reviewer is confident but not certain that the evaluation is correct

**Scope:**

4: The work is relevant to the Web and to the track, and is of broad interest to the community

---

### Official Review · Reviewer_MaFn · 2024-11-28

**Novelty:** 3
**Technical Quality:** 4

**Review:**

**Paper Summary:**

This paper introduces a self-supervised graph model that aims to balance the smoothness of graph embeddings. Experiments on real-world datasets demonstrate that the proposed approach achieves better results on benchmark datasets.

**Paper Strengths:**

1. This paper is well-organized and easy to follow.

2. This paper introduces a self-supervised graph model with theoretical guarantees.

3. The proposed method shows better performance than many baselines.

**Paper Weaknesses:**

1. The novelty of the proposed method is limited, since the constraint on neighbor-based similarity is widely used in recent literature. The authors need to highlight the contributions and novelty of this paper.

2. The method and theoretical analysis in this paper are based on the homophily assumption, which limits its applicability to heterophilic graphs—an important category in real-world scenarios. Specifically, if the graph is heterophilic, $ z_{\text{neigh}} $ may fail to capture meaningful neighborhood information. Could the authors discuss how this approach could be extended to heterophilic graphs?

3. Many other works [1] have introduced novel objective functions along with theoretical analysis. It would be helpful to include a comparison of Theorem 2 with these approaches, as well as to incorporate these methods into the experimental section for a more comprehensive evaluation.

4. The improvements observed in the experimental results are quite marginal. The authors could consider evaluating the method on additional graph-related tasks, such as graph classification or node clustering. Furthermore, it would strengthen the paper to include experimental results that validate the proposed method's ability to achieve a proper balance of smoothness, which is the core motivation of this work.

*[1] HomoGCL: Rethinking Homophily in Graph Contrastive Learning, KDD 2023*

I am very willing to increase my score on seeing a fixed revision.

**Questions:**

Please see the weaknesses above.

**Reviewer Confidence:**

4: The reviewer is certain that the evaluation is correct and very familiar with the relevant literature

**Scope:**

4: The work is relevant to the Web and to the track, and is of broad interest to the community

---

### Official Review · Reviewer_efun · 2024-12-02

**Novelty:** 4
**Technical Quality:** 3

**Review:**

**Summary**

This paper introduces BSG (Balancing Smoothness in Graph SSL), a framework designed to balance graph embedding smoothness in self-supervised learning (SSL). By decomposing the SSL objective into neighbor loss, minimal loss, and divergence loss through an information-theoretic perspective, the proposed approach aims to enhance representation quality. The authors validate their method through extensive experiments on real-world datasets, covering both node classification and link prediction tasks.

**Strengths**

1. The paper is well-written and easy to follow, addressing a relevant and necessary topic in the graph community.

2. The authors propose a new objective function to balance graph embedding smoothness.

3. Extensive experiments are conducted on multiple tasks, including node classification and link prediction. Additionally, the authors provide their implementation code for reproducibility.


**Weaknesses**

1. The paper claims that existing graph SSL methods fail to balance feature smoothing and essential graph properties effectively. However, the authors do not clearly define the "graph properties" in focus. Graphs have various properties, such as degree, adjacency, and clustering coefficient. It remains unclear which properties are specifically addressed in this work.

2. As discussed by the authors in the related work section, several existing methods aim to alleviate over-smoothing in graphs. However, the paper lacks a comprehensive experimental comparison with these methods. Additionally, the distinctions between the proposed approach and these existing methods should be explained in the section of introduction.

3. The authors claim that existing studies tend to be either over-smoothed or under-smoothed. However, as far as I know, several related studies evaluate their methods on both homophilic and heterophilic graphs, demonstrating promising performance across these scenarios.

4. In Section 5.1, the authors should categorize the baselines, clearly indicating which ones exhibit over-smoothing and which ones exhibit under-smoothing.

5. The anonymous code repository would benefit from a more detailed README to facilitate easier understanding and execution of the provided code implementation.

6. The derivation process of the formulas in Section 4.1 should be included in the appendix for clarity. Furthermore, the transition from Equation (5) to Equation (6) is unclear and requires additional explanation.

**Questions:**

Please refer to the weaknesses. If the concerns can be well addressed, I will raise the score.

**Reviewer Confidence:**

3: The reviewer is confident but not certain that the evaluation is correct

**Scope:**

4: The work is relevant to the Web and to the track, and is of broad interest to the community

---

### Official Review · Reviewer_ZQuQ · 2024-12-02

**Novelty:** 5
**Technical Quality:** 5

**Review:**

## Summary

The paper introduces BSG, a framework for balancing graph embedding smoothness in self-supervised learning (SSL) via information-theoretic decomposition. It decomposes the SSL objective into three terms: neighbor loss, minimal loss, and divergence loss, aiming to optimize representation quality in graph-based SSL. The approach supplements existing methods by considering the smoothness of node representations relative to their neighbors, crucial for capturing graph properties. Extensive experiments on real-world datasets demonstrate BSG's state-of-the-art performance in node classification and link prediction tasks.

## Pros

1. The proposed framework is highly credible and supported by rigorous theoretical analysis.
2. The comprehensive experiments provide strong evidence supporting the effectiveness of the proposed framework.

## Cons
1. The quality of the illustrations could be improved, as the current diagrams do not effectively aid in understanding the proposed framework. Additionally, the drawings appear somewhat informal, with issues such as misalignment of $Z_{s}$ and $Z_{neigh}$ at the bottom and arbitrary connections between nodes and edges. Also, there are some typos in the paper, like in Section 2.2 line 2, "more smoother"; in Table 1, A.R. column, the "4.00" is bolded.

**Questions:**

1. Does the proposed method significantly increase the training time? It would be helpful to provide an estimation of the computational complexity or experimental measurements of the training time.
2. Theorem 2 assumes the graph is homophilic, does the proposed framework remain effective when applied to heterophilic graphs?

**Reviewer Confidence:**

3: The reviewer is confident but not certain that the evaluation is correct

**Scope:**

3: The work is somewhat relevant to the Web and to the track, and is of narrow interest to a sub-community

---

### Official Review · Reviewer_p3rq · 2024-12-03

**Novelty:** 5
**Technical Quality:** 5

**Review:**

**Summary** This paper proposes BSG (Balancing Smoothness in Graph SSL) to decompose the SSL objective function into three terms via information theory. Specifically, **neighborhood loss** maximizes the mutual information between the center node and its neighbors, **minimal loss** minimizes the entropy between node information and the pretext label, and **divergence loss** maximizes the entropy between the center node and its neighbors to mitigate the oversmoothing problem in graph learning. The authors provide comprehensive theoretical and experimental analysis to support the effectiveness of the proposed method.


**Pros**:
1. Decomposing the loss function into three terms via information theory is an interesting and novel approach.
2. The authors provide a comprehensive experimental study of the model.
3. The paper is well-written and the theorems are clearly proven.

**Cons**:
1. Some results are not statistically significant in node classification (Table 1). Also, the authors do not explain the proposed method's suboptimal performance on certain datasets.


**Note**:
Please use the full name of SSL in the first line of the abstract.

**Questions:**

1. Are the objective functions for node classification and edge prediction identical? If not, could you please clarify the differences? Additionally, what are the reasons behind the model's superior performance on link prediction compared to node classification?

2. Is there any experimental evidence to support the claim that divergence loss can mitigate the oversmoothing problem?

3. What are the limitations and potential future directions of the proposed approach? Does the additional regularization increase the training time of the model?

**Reviewer Confidence:**

2: The reviewer is willing to defend the evaluation, but it is likely that the reviewer did not understand parts of the paper

**Scope:**

3: The work is somewhat relevant to the Web and to the track, and is of narrow interest to a sub-community